# Potential of the Oxidized Form of the Oleuropein Aglycon to Monitor the Oil Quality Evolution of Commercial Extra-Virgin Olive Oils

**DOI:** 10.3390/foods12152959

**Published:** 2023-08-04

**Authors:** Sonia Esposto, Stefania Urbani, Roberto Selvaggini, Agnese Taticchi, Tullia Gallina Toschi, Luigi Daidone, Alessandra Bendini, Gianluca Veneziani, Beatrice Sordini, Maurizio Servili

**Affiliations:** 1Department of Agricultural, Food and Environmental Sciences, University of Perugia, 06126 Perugia, Italy; sonia.esposto@unipg.it (S.E.); stefania.urbani@unipg.it (S.U.); agnese.taticchi@unipg.it (A.T.); luigi.daidone@unipg.it (L.D.); gianluca.veneziani@unipg.it (G.V.); beatrice.sordini@unipg.it (B.S.); maurizio.servili@unipg.it (M.S.); 2Department of Agricultural and Food Sciences, University of Bologna, 40126 Bologna, Italy; tullia.gallinatoschi@unibo.it (T.G.T.); alessandra.bendini@unibo.it (A.B.)

**Keywords:** extra-virgin olive oil quality evolution, light exposure, extra-virgin olive oil quality markers, K_270_, ΔK, (*E*, *E*)-2,4-decadienal, (*E*)-2-decenal, the oxidized form of the oleuropein aglycon (3,4-DHPEA-EA-OX)

## Abstract

The quality of commercially available extra-virgin olive oils (VOOs) of different chemical compositions was evaluated as a function of storage (12 months), simulating market storage conditions, to find reliable and early markers of the virgin olive oil (VOOs) quality status in the market. By applying a D-optimal design using the Most Descriptive Compound (MDC) algorithm, 20 virgin olive oils were selected. The initial concentrations of oleic acid, hydrophilic phenols, and α-tocopherol in the 20 VOOs ranged from 58.2 to 80.5%, 186.7 to 1003.2 mg/kg, and 170.7–300.6 mg/kg, respectively. K_270_, ∆K, (*E*, *E*)-2.4-decadienal and (*E*)-2-decenal, and the oxidative form of the oleuropein aglycon (3,4-DHPEA-EA-OX) reflected the VOO quality status well, with 3,4-DHPEA-EA-OX being the most relevant and quick index for simple monitoring of the “extra-virgin” commercial shelf-life category. Its HPLC-DAD evaluation is easy because of the different wavelength absorbances of the oxidized and non-oxidized form (3,4-DHPEA-EA), respectively, at 347 and 278 nm.

## 1. Introduction

The consumption and production of VOO have seen a huge international increase mainly because of its recognized health benefits and peculiar sensorial characteristics [1,2,3,4,5]. However, this significant rise in interest makes producers and packagers eager to im-prove olive oil quality protection during its market shelf life. The shelf life is defined as the period during which the food product will remain safe for consumption; keep the desired sensory, chemical, physical, and microbiological characteristics; will stay acceptable for consumers; and will uphold any declaration reported on the label [1].

The storage of VOOs might promote quality loss, according to the oxidative phenomena correlated to the oxygen concentration and light exposition. Several studies have demonstrated that light exposure over storage is the main factor responsible for quality loss [2,3,4,5,6,7,8].

Photo-oxidation compromises VOO’s legal, health-promoted, and sensory quality once it is marketed because of the accumulation of hydroperoxides and their rapid homo-lytic cleavage in C7-C11 volatile compounds responsible for the “rancid” defect [4,5,6,7,8]. Consequently, discrepancies could cause contradiction between the quality declared on the label, the relative consumers’ expectations, and the actual VOO quality in the market.

Indeed, suitable instruments for monitoring VOO’s shelf life must be provided considering the wide variability in terms of fatty acid composition and antioxidant content affecting the huge number of VOOs in the real market [9].

Peroxide value, extinction coefficients, volatile compounds related to the “rancid” defect and the overall and sensory assessment [4,5,6,7,8,10,11,12,13,14,15], chlorophylls, and their derivatives pyro pheophytins (PPPs) and diacylglycerols (DAGs) [16,17,18,19,20,21,22,23] have been recognized as potential markers of the quality status of VOOs.

Furthermore, since the recognized involvement of secoiridoid derivatives in the oxidative processes, new studies have focused on their oxidative products as discriminants of the actual VOO quality under market conditions.

Despina et al., monitoring the evolution of the phenolic composition in VOOs subjected to heat treatment, highlighted the accumulation of oxidized phenolic compounds in products [24]. Di Maio et al. [25,26] concluded that hydroxytyrosol (3,4-DHPEA) is involved in reactions of VOO oxidation processes and that the oxidation product of the dyaldheidic form of the elenolic acid linked to hydroxytyrosol (3,4-DHPEA-EDA or oleacein) that accumulates early in the VOO can be used as a molecular marker of the oil the VOO’s oxidation status [25,26]. Esposto et al. confirmed the constant and rapid accumulation during deep oil frying, or storage in normal conditions of the oxidized form of 3,4-DHPEA-EDA, according to the initial concentration of 3,4-DHPEA-EDA [7,27]. More recently, Di Serio et al. [28] included the oxidized phenols in the development of an analytical method to predict the remaining shelf-life of individual marketed VOOs [28]. Tsolakou et al. [29] investigated the presence of oleocanthalic acid, in VOOs stored in different conditions, observing its increase with storage time while the oleocanthal concentration decreased. The conversion of oleocanthal in oleocanthalic acid and its relative accumulation in VOOs kept at different storage conditions was also observed by Esposito Salsano et al. [15]. Abbattista et al. [30] proposed a reverse-phase liquid chromatography coupled with high-resolution/accuracy Fourier transform mass spectrometry with electrospray ionization to monitor the accumulation of oxidative forms of oleuropein and ligstroside aglycones, oleacin, and oleocanthal during storage up to 6 months under controlled conditions. Antoniadi et al. [31], after the characterization of the main oxidized olive oil secoiridoids, in several commercial Greek VOOs, hypothesized that they could contribute as chemical markers for the quality categorization of VOOs as extra-virgin, virgin, or lamp [31].

For the first time, the potentiality of the oxidized form of the oleuropein aglycon (3,4-DHPEA-EA-OX), as a new marker of the quality status of the VOO stored under market conditions, was evaluated because of the more rapid and earlier involvement of oleuropein derivatives to contrast VOO photo-oxidation [5,7,8,27,32]. A simple and clear identification method of 3,4-DHPEA-EA was also demonstrated.

## 2. Materials and Methods

### 2.1. Solvents and Reagents

For the chromatographic analysis, methyl alcohol, n-hexane, isopropyl alcohol, and glacial acetic acid, all of HPLC grade, and water, methyl alcohol, and glacial acetic acid, all of LC-MS grade, were obtained from VWR (Milan, Italy). Analytical- and HPLC-grade water was obtained using purification systems. Phenolic compounds such as (p-hydroxyphenyl) ethanol (p-HPEA) and (3,4-dihydroxyphenyl) ethanol (3,4-DHPEA), vanillic acid, caffeic acid, and α-tocopherol were supplied by Merck (Milan, Italy). The isomer of oleuropein aglycon (3,4-DHPEA-EA), the dialdehydic forms of elenolic acid linked to tyrosol and hydroxytyrosol (p-HPEA-EDA and 3,4-DHPEA-EDA, nowadays named oleconthal and oleacein, respectively), and lignans (+)-1-acetoxypinoresinol and (+)-pinoresinol were extracted from a VOO as reported by Selvaggini et al. [33]. Pure analytical standards of volatile compounds were purchased from Merck (Milan, Italy).

### 2.2. VOO Samples

Twenty-three VOOs were selected from among several VOOs purchased from Italian, EU and extra-EU providers according to their legal quality parameters (acidity, peroxide number, K_232_, K_270_, ΔK), acidic composition and antioxidant composition (α-tocopherols and total phenols). Among those 23 samples, we selected the 6 most representative VOOs (named A, C, D, E, F and G) covering the realistic quality range in the market [9]. Through a statistical set-up, 51 virtual VOOs, characterized by different contents of oleic acid, hydrophilic phenols, and α-tocopherol, were virtually created by mixing of the 6 “mother” samples at different percentage ratios (50%/50% or 25%/75%). Among the 51 virtual VOOs, according to the D-optimal design (Appendix A) using the Most Descriptive Compound (MDC) algorithm, 20 virgin olive oils, were selected because of their low, medium, and high concentrations of oleic acid, phenolic compounds, and α-tocopherol. The different percentages of the 6 “mother samples” used to realistically produce the 20 VOOs selected, are reported in Appendix A. An alphanumeric code from S1 to S20 was attributed to each VOO (Appendix A).

### 2.3. Simulation of the VOO Real-Time Shelf-Life

The twenty VOOs selected and realistically produced as described above were packaged in 500 mL UVA-grade green glass bottles, sealed with screw caps, and placed on shelves for a 12-month real-time shelf-life light exposure (500 lux for 12 h/day) at a temperature of 25 °C. The bottles were moved two times per week, from the first to the last position, into the same row, to ensure equal light exposure over the experimental period.

One bottle of each VOO was taken monthly to carry out the full analytical evaluations. Before analysis, the samples were stored at 12 °C in the darkness. All the analyses were carried out within 1 week of the VOO being withdrawn.

### 2.4. VOO Analytical Evaluations

#### 2.4.1. Regulatory Quality and Freshness Indices

The free acidity, peroxide values (PV), and extinction indices, of VOOs from T0 to T12 were determined according to the official methods of the European Commission Regulation [34]. Fatty acid (FA) composition (only for T0 samples) was estimated in accordance with the regulations of the European Union [34]. The FA and 1,2-diacylglycerol (DAG) content of the oils (only for VOOs at T0) were determined using a Dani Master GC (DANI Instruments, Milan, Italy) equipped with a flame ionization detector (FID) in accordance with the procedures described in EU Reg. 2022/2104 and ISO 29822:2009, respectively [34,35].

#### 2.4.2. Oxidative Stability Index (OSI) Time

The test to determine the OSI time (hour), was carried out only on the 20 VOOs at time 0, as follows: with a stream of purified air (120 mL min-1 air flow rate), which passed through a 5 g of sample, and the effluent air from the oil was then bubbled through a vessel containing deionized water. The effluent air contained volatile organic acids such as formic acid and other volatile compounds formed during thermal oxidation of the oil, increasing the conductivity of the water. The temperature at which this test was carried out was 110 °C. The OSI (or OSI time) was expressed in hours [36].

#### 2.4.3. α-Tocopherol

A total of 1 g of oil was dissolved in 10 mL of n-hexane, filtered with a 5-μm polyvinylidene difluoride (PVDF) syringe filter (Whatman, Clifton, NJ, USA) and injected into the HPLC system. The HPLC analysis was conducted using an Agilent Technologies Model 1100 consisting of a vacuum degasser, a quaternary pump, an autosampler, a thermostatic column compartment, a diode array detector (DAD) and a fluorescence detector (FLD). α-Tocopherol was evaluated according to the procedure of Psomiadou and Tsimidou [37] with the following modifications: n-hexane/2-propanol (99.5:0.5 *v*/*v*) (A) and n-hexane/2-propanol (70:30 *v*/*v*) (B) were used as eluents. The gradient was as follows: 100% A for 2 min, to 89% A in 8 min and maintained for 6 min, to 45% A in 2 min, then to 20% A in 2 min, maintained for 6 min, 100% A in 4 min, and maintained for 5 min. The total run time was 35 min, and the injection volume was 50 μL. α-Tocopherol was detected at an excitation wavelength of 294 nm and at an emission wavelength of 330 nm. Quantification was performed by using the curve constructed with the standard solutions of α-tocopherol. Results were expressed as mg/kg of oil.

#### 2.4.4. Polyphenols

The extraction of VOO phenolic compounds was performed in accordance with Taticchi et al. [38]. The HPLC analyses of the phenolic extracts were conducted according to Selvaggini et al. [33] with a reversed-phase column using an Agilent Technologies system Model 1100 (Agilent Technologies, Santa Clara, CA, USA), which was composed of a vacuum degasser, a quaternary pump, an autosampler, a thermostatic column compartment, a DAD, and a fluorescence detector (FLD). The C18 column used in this study was a Spherisorb ODS-1 250 mm × 4.6 mm with a particle size of 5 μm (Waters, Milford, MA, USA); the injected sample volume was 20 μL. The mobile phase was composed of 0.2% acetic acid (pH 3.1) in water (solvent A)/methanol (solvent B) at a flow rate of 1 mL/min and the gradient changed as follows: 95% A/5% B for 2 min, 75% A/25% B in 8 min, 60% A/40% B in 10 min, 50% A/50% B in 16 min, and 0% A/100% B in 14 min; this composition was maintained for 10 min and then returned to the initial conditions and equilibration in 13 min; the total running time was 73 min. The oxidized form of the oleuropein aglycon (3,4-DHPEA-OX) was detected by using the DAD set at 339 nm; all the other phenolic compounds were detected by DAD at 278 nm, the analyses and the elaboration of chromatographic data were conducted using the ChemStation software version A.10.02 (Agilent Technologies, Palo Alto, CA, USA). The quantitative evaluation of the phenols was carried out using single calibration curves for each compound, and the results are expressed as mg/kg of oil. The 3,4-DHPEA-OX was quantified using the response factor of caffeic acid, because the standard compound was not commercially available.

#### 2.4.5. Oxidized (Acidic) Form of the Oleuropein Aglycon (3,4-DHPEA-OX)

The confirmation of 3,4-DHPEA-OX was performed by UHPLC-DAD-Q-TOF/MS using an Ultra High Performance Liquid Chromatography (UHPLC) system Agilent Technologies mod. 1260 Infinity composed of a degasser, a binary pump, an autosampler, a thermostatic column compartment, and a diode array detector (DAD), coupled with a quadrupole-time of flight (Q-TOF) mass spectrometer with an electrospray ionization source (Dual ESI) Agilent mod. 6530 Accurate-Mass Q-TOF LC/MS (Agilent Technologies, Santa Clara, CA, USA). The column used was a Poroshell 120, EC-C18, 3.0 × 50 mm, 2.7 µm (Agilent Technologies, Santa Clara, CA, USA). The injected sample volume was 1 μL, and the elution was performed with a flow of 0.7 mL/min using water with 0.2% acetic acid as solvent A and methanol with 0.2% acetic acid as solvent B. The elution gradient was changed as follows: at 0 min, the solvent ratio was 85% of A and 15% of B; at 10 min, the ratio was 40% of A and 60%; at 15 min, the ratio was 100% of B, and this composition was maintained for 5 min. The system was then restored to its initial condition and allowed to equilibrate for 5 min. The total time of the analysis was 20 min. DAD spectra were acquired in the range of 190–640 nm. The MS analysis was acquired in negative mode in the *m*/*z* range of 50–1600 with a scan speed of 1 spectrum/s, and for accurate mass measurements, two reference masses (by using a calibrating solution infused through the second nebulizer) of 112.985587 and 980.016375 *m*/*z* were used. The parameters of the Dual ESI source were as follows: drying gas flow, 9 L/min; gas temperature, 325 °C; nebulizer pressure, 35 psig; capillary voltage, VCap 3500 V; fragmentor, 150 V; skimmer, 65 V, octapole 1 RF, 750 V. To confirm the structure of 3,4-DHPEA-OX a MS/MS analysis, using as precursor ion 393.1194 *m*/*z*, was always performed always in negative mode with the following parameters: VCap, 3500 V; collision energy, 15 V and scan speed of 1.5 spectrum/s, with all the other parameters unchanged. The analyses and data processing were performed using Agilent “MassHunter” software version 10.1, and for the elucidation of the structure of 3,4-DHPEA-OX, Molecular Structure Correlator (MSC) version 8.2 was used (Agilent Technologies, Santa Clara, CA, USA) (Appendix A).

#### 2.4.6. Volatile Compounds

Evaluation of volatile compounds in VOOs was performed via headspace solid-phase microextraction, followed by gas chromatography-mass spectrometry (HS-SPME/GC-MS), according to the method of Taticchi et al. [38]. A total of 3 g of oil was mixed with 2-methylpropyl acetate as internal standard at the concentration of 9.8 mg/kg. To sample the headspace volatile compounds, solid-phase microextraction (SPME) was applied as follows: all of the vials were held at 35 °C for 10 min, and then the SPME fiber (a 50/30 μm 2 cm long DVB/Carboxen/PDMS, Stableflex; Supelco, Inc., Bellefonte, PA, USA) was exposed to the vapor phase for 30 min to sample the volatile compounds. The gas chromatography mass spectrometry analyses (GC-MS) were performed using an Agilent Technologies GC 7890B with “Multimode Injector” (MMI) 7693A coupled to a single quadrupole MSD mod. 5977B using an EI Extractor (XTR) source (Agilent Technologies, Santa Clara, CA, USA); a thermostatic PAL3 RSI 120 autosampler equipped with a fiber conditioning module and an agitator (CTC Analytics AG, Zwingen, Switzerland) was also employed. The chromatographic conditions used to analyze volatile compounds were the same reported by Taticchi et al. [38]. Volatile compounds were identified by comparing their mass spectra and retention times with those of authentic reference compounds. and with spectra in the NIST 2014 mass spectra library. The quantitation of the volatile compounds was performed using the calibration curves for each compound with the internal standard method and the results are expressed as mg/kg of oil.

### 2.5. Statistical Analysis

SigmaPlot software v12.3 (Systat Software Inc., San Jose, CA, USA) for performing a priori one-way analysis of variance (ANOVA) with the Tukey test (*p* < 0.05).

The SIMCA 13.0 chemometric package was used (Umetrics AB, Umeå, Sweden) to perform multivariate statistical analyses. For this statistical elaboration, the analytical data were previously put in a matrix with the samples (n objects) in rows and the analytical parameters (k variables) in columns. The raw data were normalized, with the subtraction of the mean, and auto-scaled, dividing these results by the standard deviation. The number of significant components was found by applying cross-validation. The results of modelling are presented in graphical form.

## 3. Results and Discussion

### 3.1. Evaluation of EVOO Quality at Time T0

The initial quality indices of the 20 VOOs at the time of packaging (T0), such as free acidity; PV, K_232_ and K_270_ extinction coefficients; and FAEEs’ fatty acids composition; phenols and the oxidative form of the oleuropein aglycon; oxidative stability; and volatiles’ compositions, was evaluated and reported in the following Appendix A. The legal parameters were within the limits defined for the “extra virgin marketable” category designated by the current EU regulation [34].

Furthermore, the 20 samples had remarkably high variability (in terms of oleic acid, α-tocopherol, total phenols and volatiles), which is strongly correlated to genetic, agronomic, and technological factors, and reflected the current variability of the commercial VOOs [9].

### 3.2. Evolution of VOOs’ Overall Quality during 12 Months of Light Exposure

The whole data set of the results obtained analyzing the 20 experimental blends each month was firstly elaborated by building a PCA (Principal Component Analysis, Appendix A), showed a clear discrimination of the samples according to the time to the light exposure, along the first component (from the left to the right of the score, Appendix A).

Successively, an orthogonal-partial least square (*O*PLS, Figure 1) of latent variables analysis was carried out to observe a potential correlation between the evolution of the VOOs’ quality and the duration of storage to light exposure. The score plot (Figure 1A) confirmed the distribution of the samples according to the time of storage, with those VOOs characterized by the longest period of conservation (from 7 to 12 months), positioned to the right and to the top side of the score plot, and the VOOs with the earliest period of conservation (from 0 to 6 months), to the left and to the lower side of the score (Figure 1A). The relative loading plot (Figure 1B) highlighted that the latent dependent variable Y (time to the light exposure), was positively correlated, particularly with the extinction coefficients K_270_ and ΔK, C_7_-C_10_ aldehydes (*E*)-decenal and (*E*, *E*)-2,4-decadienal, and DHPEA-EA-OX. On the contrary, negative correlations between Y and all the antioxidant compounds, especially with the oleuropein derivatives oleacein (3,4-DHPEA-EDA) and 3,4-DHPEA-EA, were demonstrated. Furthermore, we observed that K_232_ and the ligstroside derivatives oleocanthal (*p*-HPEA-EDA) and *p*-HPEA-EA, as well as lignans, together with volatile compounds such as hexanal, nonanal, decanal and propionic acid, occupying the center of the loading plot, did not show correlation with time (Figure 1B).

This general exploration allowed us to conclude that, among all the chemical substances evaluated on the 20 VOOs over the storage to the light exposure evolution of K_270_ and ΔK, aldehydes (*E*, *E*)-2,4-decadienal and (*E*)-2-decenal, the phenolic composition and in particular the oleuropein derivatives, like 3,4-DHPEA and 3,4-DHPEA-EA-OX, were strongly correlated to the quality/freshness evolution of the VOOs.

Those data were very similar to those published by Esposto et al. (2017) [7], where PCA and PLS models showed the same positive correlations with the time to exposure to the light of different VOOs and parameters like K_270_, C_7_-C_11_ aldehydes, and the oxidative products of polyphenols [7]. However, in this research, besides K_270_, ∆K and the aldehydes (*E*, *E*)-2,4-decadienal and (*E*)-2-decenal also appeared more positively correlated to the time of conservation of the VOOs.

Using the previous separation of the 20 samples into two groups, according to the initial low-medium and high concentration of polyphenols (VOOlmp and VOOmhp groups, respectively), we followed the evolution of those parameters that more significantly influenced the quality evolution of the VOO, according to the PLS model.

### 3.3. Evolution of Extinction Indexes K_270_ and ΔK during 12 Months of Light Exposure

In this study, the evolution of K_270_ showed a general tendency to increase during the storage period (Figure 2; Appendix A) for all the samples analyzed with strong differences in terms of the number of months in which the samples remained in the extra-virgin category (limit of 0.20) [34], according to the initial concentration of polyphenols.

In fact, Figure 2A,B highlighted that in those VOOlmp, the legal limit [34] was reached by S12 and S2 within the first four months; by S11, S19, and S20 within 6 months; and by the rest of the VOOlmp samples (S13 and S17) within the 10 months of storage to the light exposure. On the contrary, S7 and S18 remained under 0.20 until the end of the experimentation.

On the contrary, in VOOsmhp (Figure 2B), only S1 exceeded the limit between the fourth and sixth month of shelf life, and only at the end of the experimentation did all the other VOOsmhp (except S4) no longer belong to the extra-virgin category [34]. These results confirmed that a lack of polyphenols promotes a quick increase in those volatile substances produced at the end of the oxidation phase, which are absorbed at 270 nm [34].

Moreover, the usefulness of K_270_ as a simple and cheap parameter for evaluating the oxidative status of a VOO [6,7,8,39,40], was confirmed.

In regard to ΔK evolution (Figure 3A,B; Appendix A) a similar tendency to increase over storage was revealed: the legal limit (0.01) for the extra-virgin category [34] was first reached by VOOlmp samples starting from the period ranging between the second and the third month (samples S8 and S11) and between the fifth and the sixth month of conservation for almost all the 10 VOOslmp, S7 and S18 samples excluded.

On the contrary, higher stability was observed in VOOmhp samples since four of ten samples (respectively S5, S4, S9, and S6) never exceeded the legal limit, whereas the other six VOOs (respectively, S1, S3, S6, S10, S14, and S15) showed value higher than 0.01 by the eighth month of storage.

However, between ΔK and K_270,_ K_270_ showed higher sensibility in revealing the oxidative status of the samples: S5, S6, and S9, which remained within the extra-virgin marketable definition according to the ΔK limit, were not according to the K_270_ parameter by the ninth month of storage.

### 3.4. Evolution of (E, E)-2,4-Decadienal and (E)-2-Decenal during 12 Months of Light Exposure

In both groups of VOOs studied, we observed a constant increase in these two substances, with differences depending on the concentration of polyphenols at time 0. The evolution of (*E, E*)-2,4-decadienal (Figure 4; Appendix A) showed that, by the fourth month of conservation of the 20 samples, a general accumulation was revealed. However, in both VOOmhp and in VOOlmp samples, the concentration stayed below the perception threshold [41] until the sixth and the tenth month of the shelf life, respectively, in VOOlmp and VOOmhp samples head spaces. Furthermore, while all VOOslmp exceeded the perception threshold [41] in VOOsmhp S1, S4, S6, S14, and S15 (50% of the group) never reached this sensory limit.

The volatile compound (*E*)-2-decenal, already revealed at time 0 in all the 20 samples, had a similar increase in (*E, E*)-2,4-decadienal and was dependent on the initial phenol concentration (Figure 5, Appendix A).

In fact, four samples belonging to the VOOslmp group (S2, S8, S11, and S12) (Figure 5A) exceeded the perception threshold (420 µg/kg) [41] by T0 within the fourth month of storage and reached a concentration of 800–1100 µg/kg at the end of the storage. However, except for S19 and S20, all the VOOslmp reached the perception threshold within 12 months (Figure 5A).

Regarding VOOsmhp, even if they exceeded the limit for storage (Figure 5B), except for two samples, S1, S3, S4, S5, and S6 stayed under the limit until the tenth month (Figure 5B). The general lower accumulation of (*E*)-2-decenal in VOOsmhp is in line with (*E*, *E*)-2,4 decadienal data, confirming that higher VOO polyphenols attenuate the accumulation of secondary volatiles products linked to the “rancid” defect [5,8,38,42,43].

### 3.5. Evolution of Polyphenols during 12 Months of Light Exposure

The evolution of all the phenolic compounds expressed as the sum of 3,4-DHPEA, *p*-HPEA, oleacein (3,4-DHPEA-EDA), 3,4-DHPEA-EA, oleocanthal (*p*-HPEA-EDA), and lignans ((+)-1-pinoresinol and (+)-acetoxypinoresnol) was also evaluated.

The data (Figure 6; Appendix A) showed a general decrease in the initial concentration, with a loss ranging between 57.1 and 86.9% of the total amount. However, while VOOslmp (Figure 6A) exhibited a depletion from 76.7 to 86.9%, in VOOsmhp, (Figure 6B), the loss was attenuated by approximately 57.1 up to 85.9%; in particular, lower decreases (57.1–65.8%) were registered in those VOOsmhp with an initial concentration of polyphenols equal to or higher than 900 mg/kg (samples S1, S4, S5, and S6) (Figure 6B).

Accordingly, Diamantakos et al. recently observed a significant decrease in phenolic content loss (46%) in Greek VOOs during usual storage with light exposure over 12 months [44].

Additionally, we showed that VOOsmhp with higher polyphenol contents were the same demonstrating higher resistance to oxidation dur to their lower values of K_270_, ΔK, (*E*, *E*)-2,4-decadienal and (*E*)-2-decenal. Thus, the antioxidant effect of VOO polyphenols, in contrast the photo-oxidation was highlighted.

These findings confirmed the results of Esposto et al. [7,8], who observed, in previous VOOs’ real-time shelf-life studies, higher resistance to oxidation in accordance with the initial polyphenol concentration [7,8].

Even Salsasno Esposito et al. [15] observed a huge decrease in oleocanthal and oleacein during the storage of 3 VOOs stored at 25 °C under daylight exposure, but at different rates: in fact, in VOOs with low polyphenol contents, oleocanthal and oleacein decreased quickly, and after only 8 months of storage, they were no longer detectable. Conversely, in the other two VOOs, containing a concentration of polyphenols, the two secoiridoids gradually decreased and were undetectable after 11 and 12 months, respectively [15].

### 3.6. Evolution of the Oxidized and Non-Oxidized Form of the Oleuropein Aglycon during 12 Months of Light Exposure

This is the first study where the 3,4-DHPEA-EA degradation product (oxidized form of oleuropein aglycon) was monitored over time to evaluate its potential capability in monitoring the oxidative status of the VOO as an earlier and reliable indicator.

In this regard, Di Maio et al. [25,26] and Esposto et al. [7] showed the accumulation of the oxidized form of 3,4-DHPEA-EDA in VOO, already demonstrating its potentiality as a marker of VOO freshness.

Similarly, Di Serio et al. [28], following the evolution of the VOOs’ quality real-time shelf life, concluded that it is also possible to predict the residual shelf-life of a VOO by measuring the relative quantity of oxidized phenols [28].

In our study, we chose to monitor only the oxidative form of the oleuropein aglycon instead of the sum of the oxidized forms or the single forms of *p*-HPEA-EDA and/or 3,4-DHPEA-EDA because the chemical structures of the last two ones usually absorb at a wavelength of 278 nm, which is the same wavelength absorbance of the relative non-acidic forms [15]. This means that spectral discrimination between the non-acidic and acidic forms, could be a long and difficult procedure, limiting the chemical evaluation as a reliable and easy method for monitoring VOO freshness over storage.

On the contrary, the discrimination between the oxidized and non-oxidized form of the oleuropein aglycon, is widely easier because of their different wavelength absorbances, respectively at 347 and 278 nm (Appendix A and Figure 7). With this regard, the lack of this substance in all 20 samples at time 0 showed the freshness of the oils at the beginning of the trial (Figure 8; Appendix A). However, its appearance in each sample started after the first month of light exposure, and after that, its accumulation differentiated according to the initial concentration of the oleuropein aglycon.

Specifically, we observed that the higher the initial level of 3,4-DHPEA-EA (Figure 9; Appendix A), the higher and faster the accumulation of the relative oxidative product (Figure 8; Appendix A). With this regard, samples of the VOOlmp group showed a slow and low accumulation of the oxidized 3,4-DHPEA-EA, which, except for sample S2, did not exceed the concentration of 30 mg/kg. On the contrary, in the VOOsmhp group, all samples reached this value, and except for S4, S9, S10, and S15, the final amounts ranged between 40 and 60 mg/kg.

The evaluation of the relative decrease in 3,4-DHPEA-EA (Figure 9; Appendix A) showed a 100% loss in all VOOslmp, whereas a residual quantity ranging between 4 and 32% was found in VOOsmhp (except for S15).

Thus, besides the total polyphenol content, it is possible to assume that the higher the initial content of the oleuropein aglycon, the higher its final residual concentration over time.

These data are in accordance with Baldioli et al. [32], who demonstrated that *o*-diphenols 3,4-DHPEA, 3,4-DHPEA-EDA and 3,4-DHPEA-EA had higher antioxidant activity than the other natural antioxidants of the VOO [32]. This finding was also confirmed by other research where oleuropein derivatives 3,4-DHPEA-EA, and 3,4-DHPEA-EDA showed higher and faster capacity in contrasting different oxidative phenomena and in preserving the oxidation of the other VOO antioxidants naturally contained, such as ligstroside derivatives, lignans, and *α*-tocopherol [7,8].

Hence, the data relative to the 3,4-DHPEA-EA evolution confirm that an estimate of the initial assets in oleuropein derivatives could be used as a predictor of the VOO shelf life in normal conditions.

## 4. Conclusions

Based on the results obtained in this research, the quality and freshness of commercial VOOs during their storage in market conditions are strongly influenced by photo-oxidation and the initial antioxidant heritage.

However, by studying the evolution of parameters that are directly involved in the modification of VOOs’ legal, health-promoted, and sensory qualities, the following was found:

K_270_ and ∆K extinction indexes and (*E*)-2-decenal and (*E*, *E*)-2,4-decadienal may be useful for monitoring the quality of stored VOOs by determining the time at which they will lose their “extra” status.

The rapid increase in the 3,4-DHPEA-EA-OX contents in the VOOs indicated the immediate and primary involvement of the oleuropein derivatives in inhibiting the oxidation phase.

The different wave lengths of absorbance of 3,4-DHPEA-EA and 3,4-DHPEA-EA-OX simplify the analytical determination of 3,4-DHPEA-EA-OX.

The results of this study clearly demonstrate the effectiveness of 3,4-DHPEA-EA-OX as a faster indicator of VOOs’ quality and freshness for the first time.

## Figures and Tables

**Figure 1 foods-12-02959-f001:**
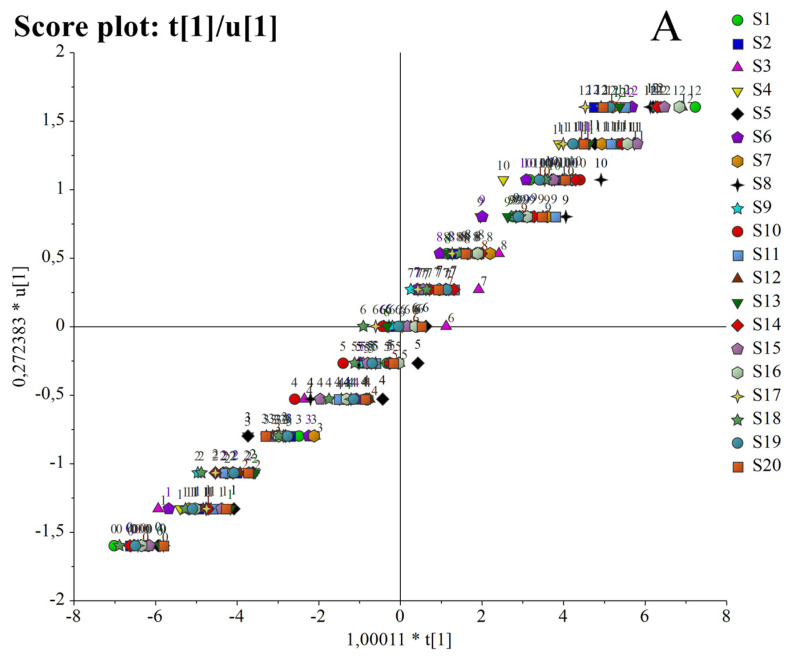
Score plot (t1/u1) (**A**) and loading plot (**B**) (pq1/poso1) of the first latent variable of the OPLS model built with all the analytical evaluations (independent variable X) for the 20 VOOs exposed to light for 12 months (latent variable Y representing the TIME). The model, explaining 61% of the total variance of X (R2X) with five components (1 predictive and 4 orthogonal, each explaining 24%, 14%, 12%, 7%, and 4%, respectively) and 98% of the overall explained variance (R2) by the model.

**Figure 2 foods-12-02959-f002:**
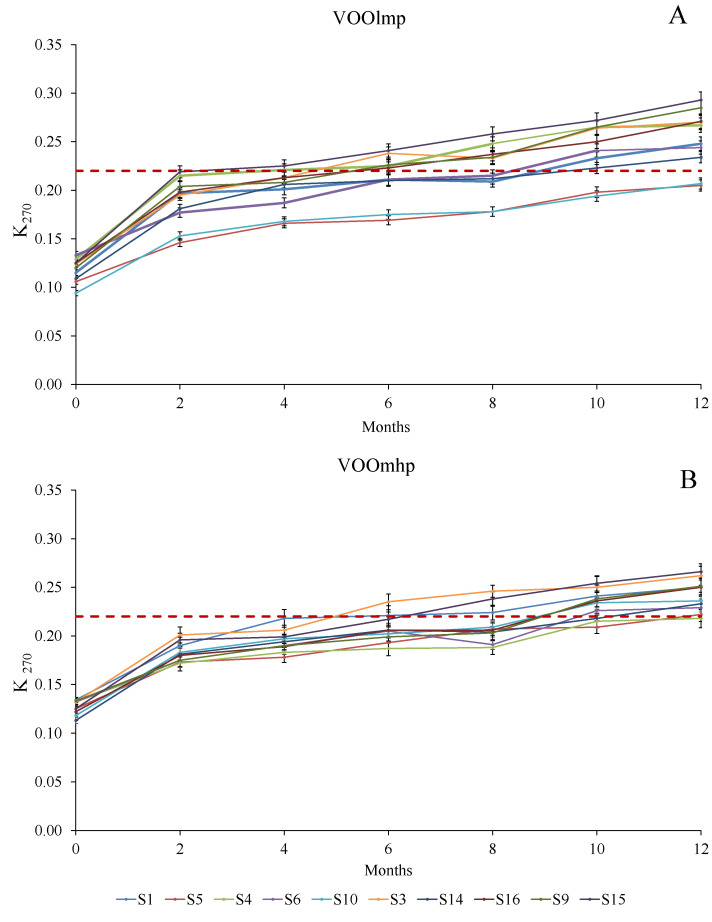
Evolution (reported every two months) of K_270_ over 12-month storage with light exposure in VOOlmp (**A**) and VOOmhp (**B**) samples. The dashed red line indicates the K_270_ legal limit (0.20) for the extra-virgin olive oil category [34]. Legend: VOOlmp: Virgin olive oils with low-medium polyphenol content; VOOmhp: Virgin olive oils with medium-high polyphenol content.

**Figure 3 foods-12-02959-f003:**
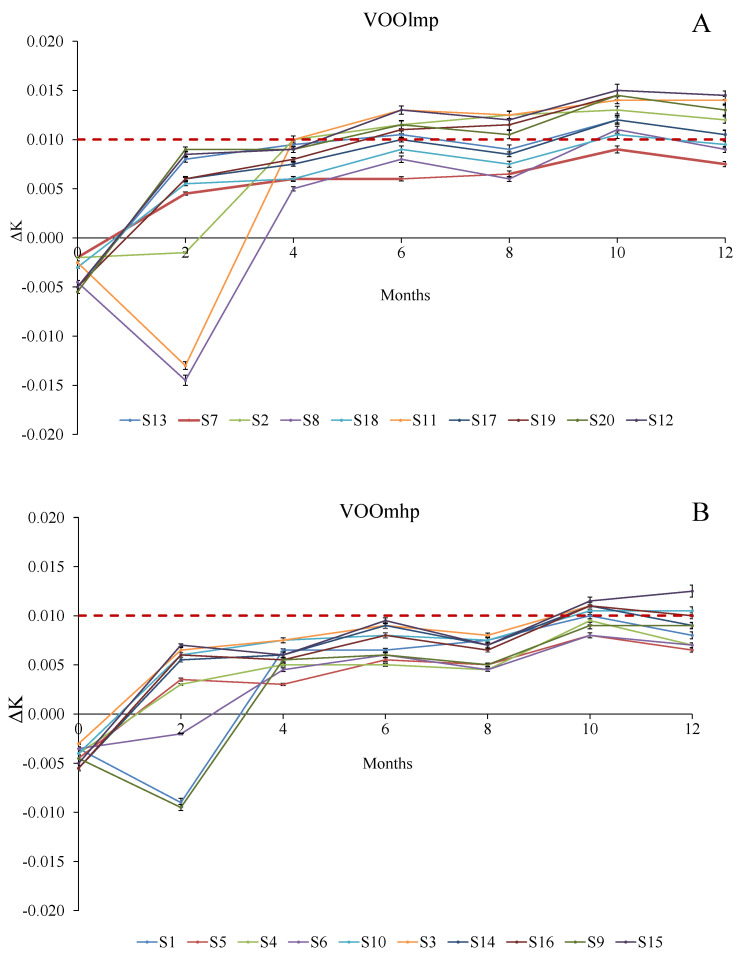
Evolution (reported every two months) of ∆K over 12-month storage with light exposure in VOOlmp (**A**) and VOOmhp (**B**) samples. The dashed red line indicates the ΔK legal limit (0.01) for the extra-virgin olive oil category [34]. Legend: VOOlmp: Virgin olive oils with low-medium polyphenol content; VOOmhp: Virgin olive oils with medium-high polyphenol content.

**Figure 4 foods-12-02959-f004:**
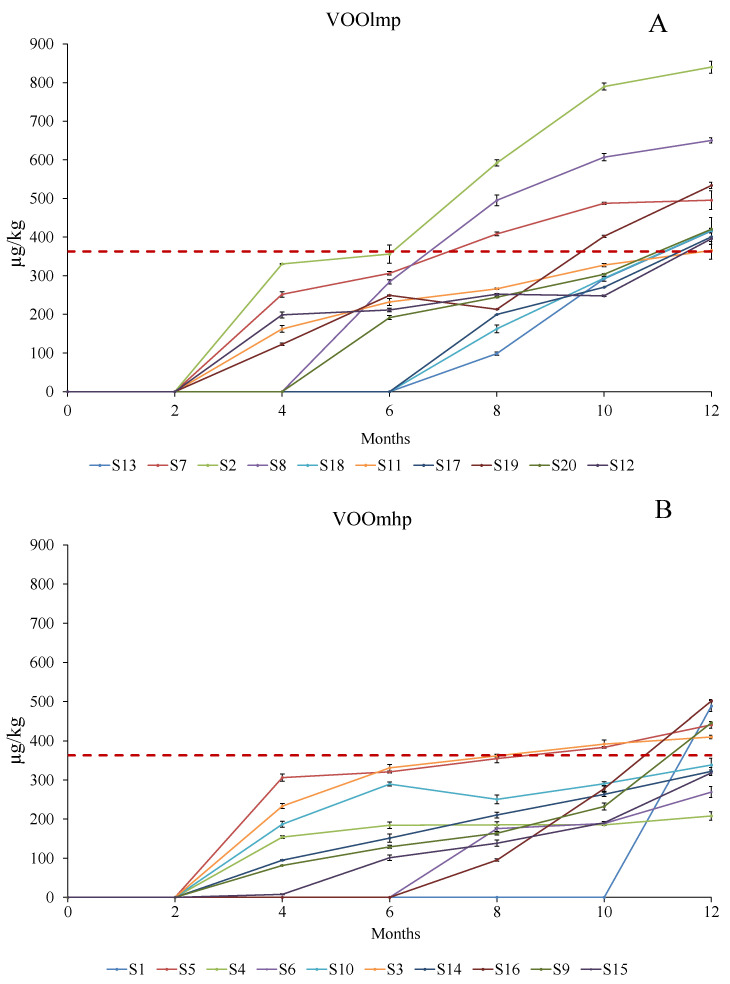
Evolution (reported every two months) of (*E, E*)-2,4-decadienal (µg/kg) over 12-month storage with light exposure in VOOlmp (**A**) and VOOmhp (**B**) samples. The dashed red line indicates the concentration representing the perception threshold (363 µg/kg) [41]. Legend VOOlmp: Virgin olive oil with low-medium polyphenol content; VOOmhp: Virgin olive oil with medium-high polyphenol content.

**Figure 5 foods-12-02959-f005:**
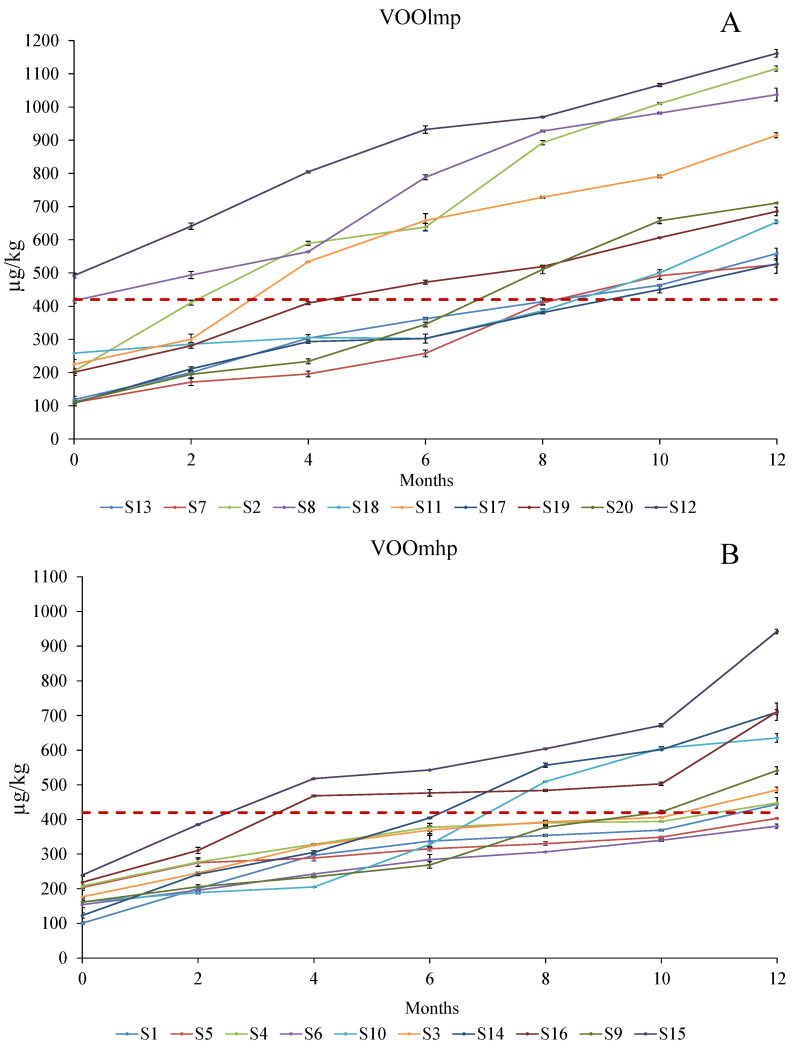
Evolution (reported every two months) of (*E*)-2-decenal (µg/kg) over 12-month storage with light exposure in VOOlmp (**A**) and VOOmhp (**B**) samples. The dashed red line indicates the concentration representing the perception threshold (420 µg/kg) [41]. Legend VOOlmp: Virgin olive oil with lowmedium polyphenol content; VOOmhp: Virgin olive oil with medium-high polyphenol content.

**Figure 6 foods-12-02959-f006:**
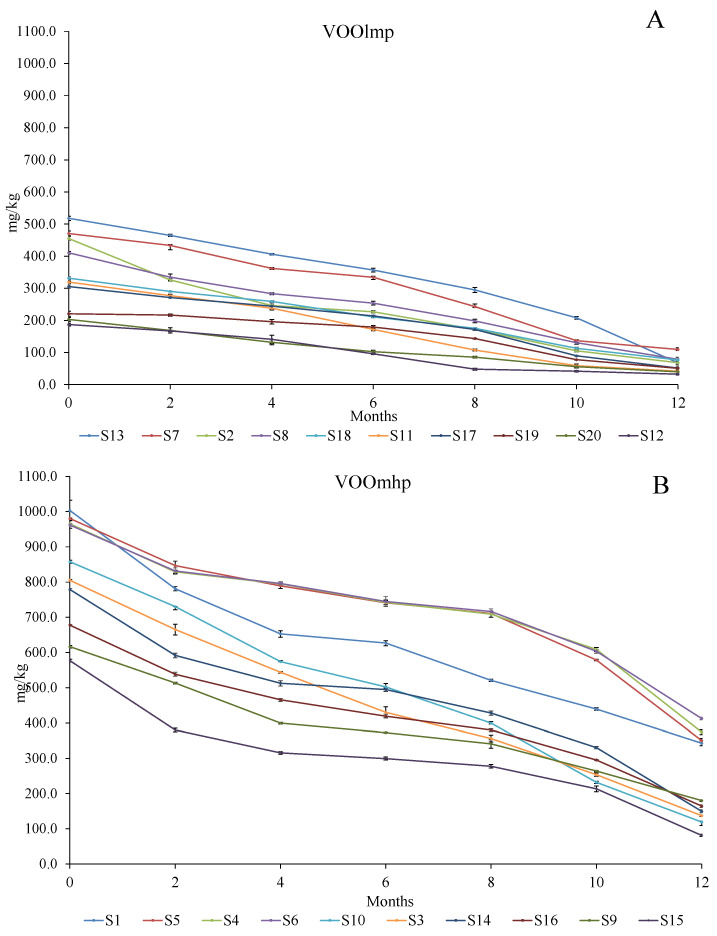
Evolution (reported every two months) of the polyphenol concentrations (mg/kg) over 12-month storage with light exposure in VOOlmp (**A**) and VOOmhp (**B**). Legend: VOOlmp: Virgin olive oil with low-medium polyphenol content; VOOmhp: Virgin olive oil with medium-high polyphenol content. The values are expressed as the sum of hydroxytyrosol (3,4-DHPEA), tyrosol (p-HPEA), oleacein (3,4-DHPEA-EDA), oleuropein aglycon (3,4-DHPEA-EA), oleochantal (p-HPEA-EDA), ligstroside aglycon (p-HPEA-EA) and lignans ((+)-1-acetoxypinoresinol and (+)-pinoresinol).

**Figure 7 foods-12-02959-f007:**
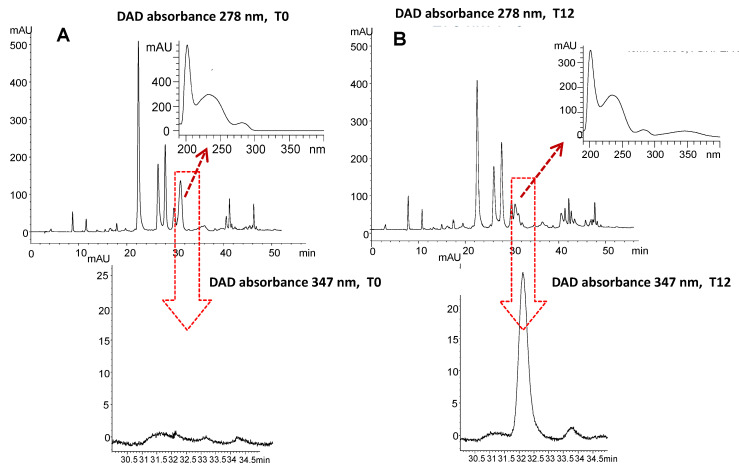
HPLC chromatogram and relative absorbance spectrum at 278 nm and 347 nm of oleuropein aglycon (3,4-DHPEA-EA) (**A**), and HPLC chromatogram and relative absorbance spectrum at 278 nm and 347 nm of the oxidized form of oleuropein aglycon (3,4-DHPEA-EA-OX) (**B**), contained in of one of the 20 VOOs, at time 0 and at time 12.

**Figure 8 foods-12-02959-f008:**
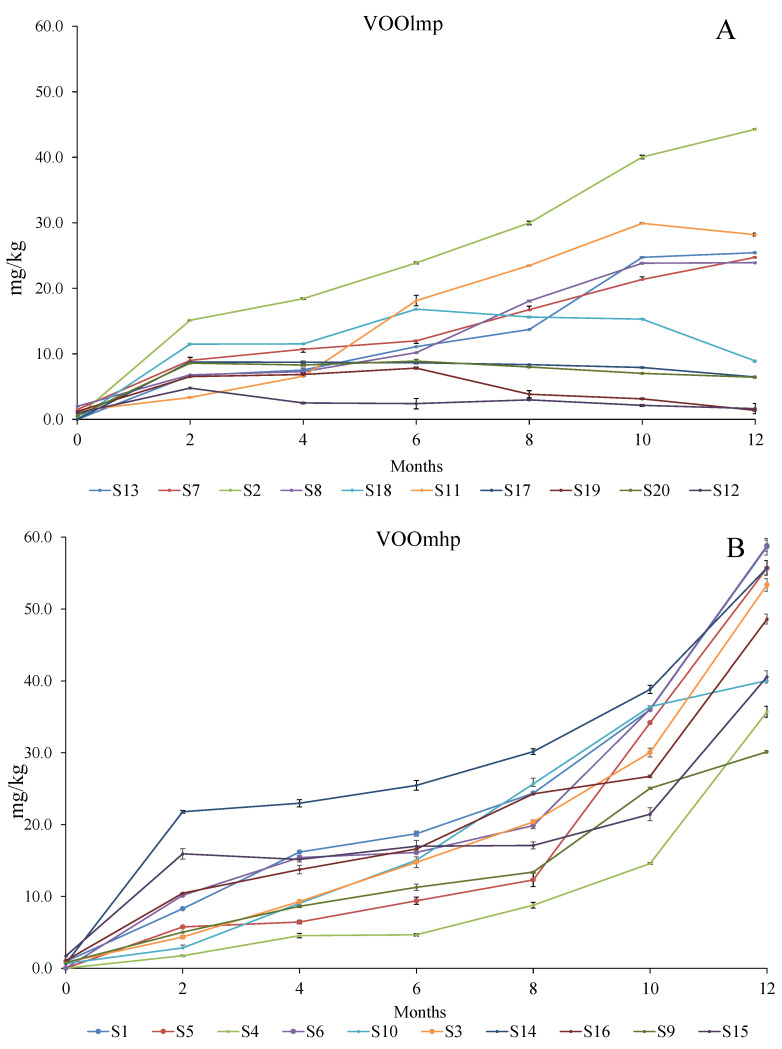
Evolution (reported every two months) of the oxidized form of the oleuropein aglycon concentration over 12-month storage with light exposure in VOOlmp (**A**) and VOOmhp (**B**). VOOlmp: Virgin olive oil with low-medium polyphenol content; VOOmhp: Virgin olive oil with medium-high polyphenol content.

**Figure 9 foods-12-02959-f009:**
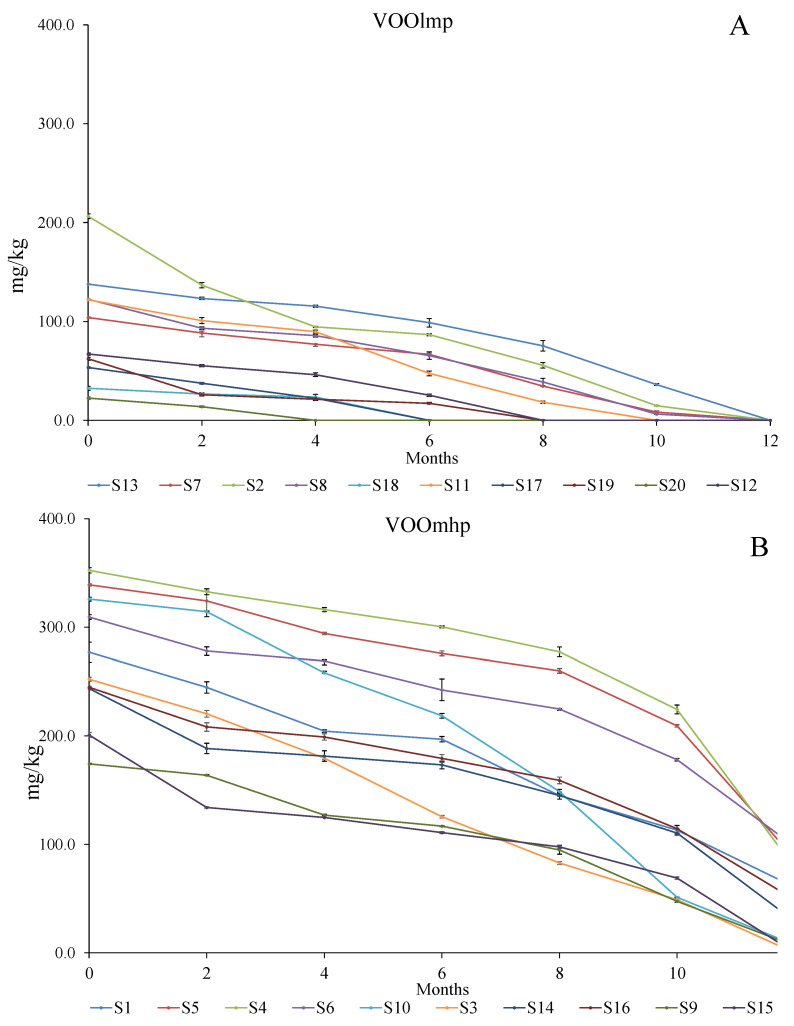
Evolution (reported every two months) of the oleuropein aglycon concentration over 12-month storage with light exposure VOOlmp (**A**) and VOOmhp (**B**). VOOlmp: Virgin olive oil with low-medium polyphenol content; VOOmhp: Virgin olive oil with medium-high polyphenol content.

## Data Availability

Data is contained within the article or Appendix A.

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
