# Peer review of "Potential of the Oxidized Form of the Oleuropein Aglycon to Monitor the Oil Quality Evolution of Commercial Extra-Virgin Olive Oils"

_foods, 2023, doi:10.3390/foods12152959_

Round 1

Reviewer 1 Report

Please see the detail comments in the attachment file.

need to improve

Reviewer 2 Report

Manuscript foods 2486512

The study focused on evaluating the oxidised form of oleuropein aglycon content. However, the evolution of many other compounds was also followed in parallel. This accumulation of information makes it difficult to follow the explanations. Besides, usually, the manuscript describes changes in full detail. Overall, the manuscript includes much information, but its understanding is complex. Among the problems observed are a)lack of concretion in the manuscript, b) unclear sentences because of faulty English construction, c)excessive detail in the evolution description, d) tables with many but hardly visible data, etc.

I think the authors did a great job, but have somewhat failed in exposing the results. Authors should revise the manuscript removing some preliminary information (e.g., information on the "mother" samples, which only were used for obtaining adequate combinations), focusing the explanations on those compounds they consider relevant, and paying attention to trends more than details. Essentially, be concrete and clear.

Additionally, some specific comments follow.   

Title. It could be simplified.

Abstract. The redaction could be improved, simplified and made more concrete.

Introduction. Could be reduced

Pg 1 L16. VOOS, Please provide full name first-time mention.

P1L24. least?

Table S1 and others. What is the statistical meaning of mean ± standard deviation?

 Including sd in parenthesis could be more realistic, just as an indication of dispersion.

Please, revise the number of mean figures retained in tables. In many cases, they exceed those significant according to the sd.  

Homogenise decimal numbers between means and sd.

Besides, in the case of so many supplementary materials, it could be interesting to facilitate the reader's understanding of the legends in each table and figure included in the supplementary material. Reading them in the manuscript and interpreting the tables and figures can result in a challenging exercise.

P7L303. Table 1. It is supposed to be Table S1. Please revise assignations throughout the manuscript. Moreover, There is confusion regarding the type of samples you refers to (6 or 20)

P10L386. All the analytical evaluations…... Please concrete.

Fig 3 and others. Variability is hardly observed. Maybe advice that they could be within the size of symbols could be convenient.  

Results

Excessive details for describing the evolutions.  

Conclusion. It could be reduced and focused on the relevant findings.

The manuscript requires improvement.

Reviewer 3 Report

The authors propose a new photooxidation marker for the distinction of fresh VOO.

The authors have designed a real shelf-life study using 20 experimental VOO to monitor during 12 months the evolution of several classical parameters, complementing with 3,4-DHPEA-EA-OX. For this last compound, two sets of VOO were defined, with different phenolic contents. The authors verified that the formation of this oxidation compound occurs under light exposure over time, and that its content is proportional to the original t 3,4-DHPEA-EA amounts.

However, it is not clear how can this quantification be of interest to shelf life studies because its study on bottling date will not reveal the outcomes of degradation, only a repeated analysis over time will. Also, both sets had similar relative degradation over a 12 month period. Indeed, a VOO on the market with a small amount of 3,4-DHPEA-EA-OX can derive from both a fresher VOO or a VOO with an initial reduced amount of 3,4-DHPEA-EA. Indeed, the estimation can only occur if, besides evaluation the 3,4-DHPEA-EA-OX during storage, the original content if known and acompaigned under similar condition to those on the market shelves. The possibility to correlate the OX amounts with other parameters, using as an indirect way to predict the VOO cold be further developed but not such approach was made despite to richness of data presents as suplemmentary table. 

The potential predictive tool should be better explained and compared with quality cut-offs for a true potential sheld-life prediction tool.

some details:

- sampling - l 173 - for a reader that starts at this reading point, it is not perceived that the 20 VOO were designed blends. The designation should be more clear throught the manuscript. the same applies to line 306 were the term samples should be clarified.

l301-304 - the % of linoleic acid is inverted between samples.

during all data discussion, it is not clear why the samples were divided in two groups based on their phenolic content since the evolution patters are quite similar and not statistical treatments between groups is shown.

correct VOOsmlp

The language is very hard to read in some +arts, particularly in the introductions, with incomplete sentences or wrong use of terms.
